# Aluminum Melt Degassing Process Evaluation Depending on the Design and the Degree of the FDU Unit Graphite Rotor Wear

**DOI:** 10.3390/ma15144924

**Published:** 2022-07-15

**Authors:** Martin Podaril, Tomáš Prášil, Jan Majernik, Rudolf Kampf, Ladislav Socha, Karel Gryc, Martin Gráf

**Affiliations:** 1Institute of Technology and Business in České Budějovice, Faculty of Technology, Okružní 517/10, 370 01 České Budějovice, Czech Republic; podaril@mail.vstecb.cz (M.P.); kampf@mail.vstecb.cz (R.K.); 2MOTOR JIKOV Slévárna a.s., Kněžskodvorská 2277, 370 04 České Budějovice, Czech Republic; tprasil@mjgroup.cz (T.P.); mgraf@mjsl.cz (M.G.); 3Faculty of Mechanical Engineering, University of West Bohemia, Univerzitní 2732, 301 00 Plzeň, Czech Republic; 4Environmental Research Department, Institute of Technology and Business in České Budějovice, Okružní 517/10, 370 01 České Budějovice, Czech Republic; socha@mail.vstecb.cz (L.S.); gryc@mail.vstecb.cz (K.G.)

**Keywords:** aluminum alloys, refining, melt degassing, FDU unit, graphite rotor

## Abstract

One of the most important indicators of casting quality is porosity. The formation of pores is largely conditioned by the presence of hydrogen in the batch and subsequently in the melt. The gasification of the melt is the primary factor increasing the porosity of casts. This paper addresses the issue of reducing the melt gasification by using FDU (Foundry Degassing Unit) unit. The gas content in the melt is evaluated by determining the Dichte Index depending on the geometry and the degree of the FDU unit rotor wear. For experiments performed under the operating conditions, three types of graphite rotors with different geometries are used. The extent of melt gasification and the Dichte Index are monitored during the rotor wear, at a rate of 0%, 25%, 50%, 75% and 100% rotor wear. Secondly, the chemical composition of the melt is monitored depending on the design and wear of the rotor. It is proven that the design and the degree of rotor wear do not have significant effect on the chemical composition of the melt and all evaluated samples fell within the prescribed quality in accordance with EN 1706. With regard to the overall comparison of the geometry and wear of individual rotor types, it has been proven that, in terms of efficiency, the individual rotors are mutually equivalent and meet the requirements for melt degassing throughout the service life.

## 1. Introduction

Among foundry technologies in the casting of aluminum alloys for the automotive industry, high-pressure die casting (HPDC) technology is at the forefront. This was achieved mainly due to little or no finishing operations in the production of parts [1]. Die-casts are characterized by high geometric accuracy, good mechanical properties and low price. Good mechanical properties of casts are related to their fine-grained structure, caused by subcooling of the melt in contact with the relatively cold wall of the mold cavity [2], but defects of casts related to porosity and the presence of gases in the melt significantly affect the quality of casts [3,4].

Due to relatively high porosity of the casts and the different proportion of gas entrapment in the cavities, arising either by shrinkage or by entrapping gases in the melt volume, the identification of the porosity type in HPDC casts is difficult [4,5,6]. In general, the presence of pores in the volume of the casts is attributed to the change in state of the substances (liquid/solid). These cavities, ranging in size from microns to millimeters, depend on the type of metal alloy and the solidification process of the casts. The formation of cavities in the structure of casts generally originates in the shrinkage of the melt during solidification, in distribution of gases contained directly in the melt, or in a combination of these factors, including gas porosity caused by gas entrapment during the casting cycle [7,8,9].

In order to obtain high-quality casts, it is necessary to comply with several parameters during the entire casting process. The HPDC process must be comprehended as a mutually correlating mechanism of die casting machine design, mold design and setting of casting technological parameters [10,11].

Regardless of the casting cycle mastering, the quality of the cast is primarily dependent on the quality of the input raw material, its melting and subsequent metallurgical treatments. To enable the production to be profitable, lower quality input materials are often used today, which, although they have the required chemical composition, may contain a relatively large number of impurities, originating not only from the original input materials, but also from their processing [12,13]. Impurities entering the process during remelting and casting are one of the primary causes of foundry defects and have negative impact on the mechanical properties of casted Al-Si components [14,15]. Solid impurities and various exotic inclusions are one of the primary impurities introduced in the process of implementation of the return material into the melt. These enter the process during the handling of return material [16,17].

Hydrogen is a major contributor to microporosity formation in aluminum alloys. The solubility of hydrogen in aluminum is significantly lower in the solid than in the liquid state. For this reason, it is generally assumed that during solidification, hydrogen is excreted from the solution and forms hydrogen pores in the form of molecular gas H_2_ [18]. Hydrogen pores cannot be formed by homogeneous or heterogeneous nucleation during solidification of aluminum alloys. During pore formation, nucleation is bypassed and pores grow simply by inflating bifilm defects. For this reason, the dissolved hydrogen content should only be controlled when bifilms are present in the liquid alloy, which is, unfortunately, a common circumstance in industrial practice [19,20].

In order to reduce the risk of casting porosity, enormous efforts have been made in the past to develop degassing melt processing technologies. The main effect of these treatments is to reduce the dissolved hydrogen content in liquid alloys. Rotary degassing is one of the most commonly used melt technologies in the foundry industry. This method involves injecting a rinsing gas into a liquid alloy through a rotating impeller [21,22]. Currently, the rinsing gas is usually nitrogen or argon; however, active gases such as chlorine, freon, fluorine or mixtures thereof with nitrogen or argon have also been used in the past [23]. Rotary degassing provides smaller and more evenly distributed rinsing gas bubbles, which is advantageous in terms of removing dissolved hydrogen as well as removing inclusions. The treatment is often combined with the application of fluxes, which can significantly increase the efficiency of removing inclusions. On the other hand, the process window, in which the processing produces a better melt quality, is small and particularly sensitive to small changes in process parameters. When fluxes are not used, the operating parameters of rotary degassing units are usually optimized for efficient removal of dissolved hydrogen [23,24,25,26]. These parameters include (and are not limited to) rotor geometry, retaining vessel and rotor dimensions, amount of metal, rinsing gas flow rate, impeller speed, treatment time, use of deflectors, initial and planned volume of hydrogen, melt temperature, chemical composition of melt and other environmental factors (such as atmospheric humidity and gas pipelines used) [27,28].

The submitted article addresses the issue of the melt degassing efficiency with respect to the rotor shape of the FDU unit and at the same time the degree of the rotor wear. Secondary to the rotor shape and rotor wear, the chemical composition of the melt is also evaluated. The actual solution of the selected issue was performed by experiments in technological procedures of melt preparation for conditions of high pressure die casting of AlSi9Cu3(Fe) alloy. The influence of the geometry of three different graphite rotors was studied. The melt gasification was evaluated by the Dichte Index and the chemical composition of the melt was monitored by optical emission spectrometry. It is proved that depending on the assessed parameters the melt is not subject to extreme changes in chemical composition and all values of chemical composition assessed for variable geometry and rotor wear fall within the prescribed melt quality specified in EN 1706. When assessing melt gasification, the phenomenon of an increased volume of gases in the melt using new, unworn rotors is evident.

## 2. Materials and Methods

As a part of evaluation of the degassing process of aluminum melt depending on the shape and degree of FDU unit graphite rotor, a series of operational experiments were performed during the refining of the alloy EN AC-46000 (AlSi9Cu3(Fe)). The melt refining process involved the preparation of a melt for the production of high-pressure casts for the automotive industry.

EN AC-46000 alloy has a good run-in attribute and a low tendency to form concentrated shrinkage. Prescribed chemical composition of the alloy is presented in Table 1.

The EN AC-46000 alloy processing procedure for HPDC technology is shown in Figure 1.

The liquid metal is produced in the STRIKO MH II-T 2000/1000 melting gas furnace for high pressure die casting technology. The furnace is charged by a loading device and the ratio of the charge during melting of the material can be in the proportion of at most 50% of the return material and at least 50% of the primary material. The material is melted at 720 °C within ±10 °C. A preheated ladle is used to transport the liquid metal. The liquid metal is poured into about 1/3 of the ladle and the refining flux is poured onto the surface of the melt in an amount of about 0.4 kg. The ladle is then filled up with molten material from a melting furnace.

The next treatment step is refining using a foundry degassing unit (FDU), in which the carrier inert gas (nitrogen under operating conditions) is being injected into the melt by a rotor. Very small gas bubbles and intensive mixing of the melt ensure fast and efficient degassing, removal of impurities and reduction in non-metallic inclusions. The melt treatment is performed on the FDU ROTO-STATIV 1-5201 device from the FOSECO company.

Technological parameters of the melt refining process for high pressure die casting on the FDU ROTO-STATIV 1-5201 (hereinafter FDU) are stated in Table 2.

Increased attention is devoted to the removal of moisture from new, unused graphite components (including rotor and shaft) that come into contact with the melt. It is assumed that unused graphite rotor and shaft contain moisture that would be transferred to the melt and thus increase the level of gasification of the melt. The removal of moisture contained in the graphite components was carried out as follows:heating of graphite components—immersion of the rotor with the bottom area to a surface whose temperature exceeded 300 °C for a period of 6 min;immersion to working height—during heating there was a significant evolution of gas manifested by bubbling. This heating by a melt was performed for 7 min;removal from melt—heated components were removed from the melt followed by launching the refining duty cycle with duration of 180 s.

This procedure of heating graphite components was realized for all three types of unused rotors.

Three types of graphite-based rotors were assessed within the operational experiments. The advantage of rotors made of graphite is lower price, higher flexibility in terms of choosing the design of rotor head (production by machining on a CNC machine), high strength and resistance to thermal shocks. The disadvantage of graphite rotors is their lower service life caused by oxidation at higher temperatures (graphite oxidizes from about 500 °C) and also by abrasion. The peeling of graphite particles from the rotor has a negative effect on the purity of the aluminum melt after refining. The loss of graphite mass is also negatively affected by the fact that as the number of duty cycles rises, the design of the refining rotor changes due to abrasion and oxidation.

Table 3, Table 4 and Table 5 are stating the settings of technological parameters. Figure 2 illustrates the geometry of individual rotors used in operational experiments.

The refining process and the lifespan of rotors were assessed according to the number of cycles at which a certain percentage of rotor wear is reached. Reaching the individual stages, or specific degree of rotor wear, was determined after consultation with the suppliers of the individual rotors. Table 6 defines the number of cycles for which the monitored rotor wear is considered. For each type of rotor, 5 series of operational experiments were performed to monitor the degree of wear 0–100%.

## 3. Results

The results obtained by the experiments can be divided into three parts. The first part describes the rotors’ wear depending on the number of cycles performed. Subsequently, the analysis of the chemical composition of the melt is performed depending on the change of the monitored parameters. The assessment of refining efficiency as a function of the gasification degree of the melt is evaluated using the Dichte Index.

### 3.1. Rotors Wear Assessment

Rotors wear and the shape degradation were monitored after completing refining cycles to the extent shown in Table 6. Figure 3, Figure 4 and Figure 5 illustrate the rotor wear at the number of duty cycles listed in Table 6.

According to Figure 3, Figure 4 and Figure 5, it is clear that due to abrasion and oxidation, the geometry of the refining rotor changes over the lifespan of the rotor and with an increasing number of duty cycles, which is primarily caused by the loss of graphite mass.

### 3.2. Chemical Composition Analysis

Chemical composition analysis of the melt was performed using Q4 TASMAN optical emission spectrometer. The values obtained were compared with the chemical composition of the melt prescribed according to EN 1706, which is stated in Table 1. Samples for optical spectrometry were taken from each ladle. Table 7, Table 8 and Table 9 show the average values of the chemical composition of taken samples of EN AC-46000 alloy.

Based on achieved results of the chemical composition analysis, which are shown on Table 7, Table 8 and Table 9, it emerges that the chemical composition of the melt corresponds to the prescribed quality according to EN 1706. Purchased and return material of the given quality are used in the production of EN AC-46000 alloy casts. It can be stated from the results that the melt was not contaminated with another alloy and complies with the standard.

### 3.3. Melt Gasification Analysis

Within the melt refining efficiency analysis, an evaluation of the melt quality was performed before and after refining at the FDU unit (sampling FDU_start_ a FDU_end_). Within the experiment, Dichte Index (DI) values were determined from the results of individual melts.

#### 3.3.1. Degassing Efficiency Evaluation Using Type A Rotor

The data measured during the operational experiments were subjected to an analysis aimed at evaluation of Dichte Index (DI). Achieved Dichte Index values before (FDU_start_) and after the melt refining process (FDU_end_) using type A rotor are statistically processed in Table 10.

From the above values, it is clear that the melt contains an increased amount of hydrogen before refining. This is evidenced by the higher expected value of DI at FDU_start_, which range from 10.97% to 8.96%. During refining process, the hydrogen content in the melt decreased, which was reflected in a decrease in expected values of DI in the range of 3.23% to 0.38%. The data show that in the case of series with 0% rotor wear, there is less significant decrease in DI values after refining process (FDU_end_). The higher DI values in this series of experiments can be attributed to the presence of rotor moisture, which is released into the melt at higher temperatures and increases the hydrogen content.

Figure 6 illustrates the representative samples for assessing the gasification degree for TYPE A rotor depending on the rotor wear degree.

#### 3.3.2. Degassing Efficiency Evaluation Using Type B Rotor

The data measured during the operational experiments were subjected to an analysis aimed at evaluation of Dichte Index (DI). Achieved Dichte Index values before (FDU_start_) and after the melt refining process (FDU_end_) using type B rotor are statistically processed in Table 11.

From the above values, it is clear that the melt contains an increased amount of hydrogen before refining. This is evidenced by the higher expected values of DI at FDU_start_, which range from 11.44 to 10.16%. During the refining process, the hydrogen content in the melt decreased, which was reflected in a decrease in expected values of DI in the range of 4.25 to 0.67%. The data show that in the case of series with 0% rotor wear, there is less significant decrease in DI values after the refining process (FDU_end_). The higher DI values in this series of experiments can be attributed to the presence of rotor moisture, which is released into the melt at higher temperatures and increases the hydrogen content.

Figure 7 illustrates the representative samples for assessing the gasification degree for TYPE B rotor depending on the rotor wear degree.

#### 3.3.3. Degassing Efficiency Evaluation Using Type C Rotor

The data measured during the operational experiments were subjected to an analysis aimed at evaluation of Dichte Index (DI). Achieved Dichte Index values before (FDU_start_) and after the melt refining process (FDU_end_) using type C rotor are statistically processed in Table 12.

From the above values, it is clear that the melt contains an increased amount of hydrogen before refining. This is evidenced by the higher expected value of DI at FDU_start_, which range from 10.94 to 9.29%. During refining process, the hydrogen content in the melt decreased, which was reflected in a decrease in expected values of DI in the range of 3.08 to 0.59%. The data show that in the case of series with 0% rotor wear, there is less significant decrease in DI values after refining process (FDU_end_). The higher DI values in this series of experiments can be attributed to the presence of rotor moisture, which is released into the melt at higher temperatures and increases the hydrogen content.

Figure 8 illustrates the representative samples for assessing the gasification degree for TYPE C rotor depending on the rotor wear degree.

## 4. Discussion

The experiments described above and the data obtained from them were focused on assessing the degassing efficiency of aluminum alloy depending on the geometry and degree of FDU unit rotor wear. At the same time, the change in the chemical composition of the alloy was monitored in accordance with EN 1706 standard.

The results of the chemical composition analysis show that the chemical composition according to the measured values corresponded to the prescribed standard EN 1706. It should be noted that in the production of AlSi9Cu3(Fe) alloy, the material used is divided into purchased and return material of the given quality. The results show that the melt was not contaminated with another alloy and complied with the standard.

To assess the efficiency of melt degassing, an assessment was chosen by comparing and evaluating the Dichte Index before and after melt refining process using Type A, Type B and Type C rotors in a wear degree of 0% up to 100%. Figure 8, which represents the percentage reduction in the Dichte Index after refining process (FDU_end_) relative to its initial value (FDU_start_), is also based on these values. For the purposes of this work, the above-mentioned percentage value was defined as the efficiency of aluminum melt refining process on the FDU unit.

It is clear from Figure 9 that the lowest DI final values are reached by the TYPE A rotor. This rotor has a refining efficiency of over 90% in the case of a series of melts with 25 up to 100% wear. During the series with 0% wear, Type A rotor achieved an efficiency of 63.9%. This low value was caused by the already mentioned moisture of the new rotor, thanks to which hydrogen is released into the melt during refining process. The same statement is valid for Type B and Type C rotors. Across the individual series of Type A rotor, and interesting trend in the achieved DI values after refining process manifested—with increasing the wear degree, the efficiency of the aluminum melt refining process increased. This trend has not manifested itself to such an extent in other rotors. However, it is necessary to take into account the fact that in the case of Type A rotor a smaller number of melts was performed, which could slightly distort the evaluation. There was also observed an increase in efficiency at 100% wear of Type B rotor (about 8% compared to 75% wear).

Despite the differences described above, it can be stated the Type A, Type B and Type C rotors achieve very similar results, as can be seen in Figure 9. The differences in the achieved values of monitored quantities are small and ultimately negligible. All rotors showed a degassing efficiency of about 90%. All rotors were also able to degas the aluminum melt to the required level as the service lifespan decreased. From this point of view, the rotors can be considered equivalent.

## 5. Conclusions

As a part of the solution of experiments monitoring the efficiency of the melt degassing on the FDU unit depending on the geometry and rotor wear degree, operational melts were performed with a focus on high pressure die casting technology and refining process of ROCKER COVER aluminum cast made of AlSi9Cu3(Fe) alloy. The samples examined were taken from melts at individually selected rotor wear intervals. Operational experiments focused on the chemical composition and gasification of the melt were performed on individual samples. The main conclusions drawn from the data obtained in this experimental research can be summarized as follows:The results of the chemical composition analysis show that the chemical composition of the individual melts complies with the EN 1706 standard. It is therefore possible to conclude that the handling of the return material that is delivered to the batch, as well as that the advancing abrasive rotor wear does not contaminate the melt and thus significantly change its chemical composition;Within the evaluation of the Dichte Index, moisture problem was found when using new graphite components at the rate of 0% wear, which caused insufficient degassing of unwanted gases from the melt. As a result, the melt did not reach the specified quality limit, which had negative effect on the monitored parameters. The moisture problem of graphite components exhibiting 0% wear can be minimized by increased attention to the preheating. To reduce humidity, it is therefore advisable to preheat new unworn rotors for at least 5 min to a temperature with a value of around 50% of the melting temperature;The other wear series of 25%, 50%, 75% and 100% reached the values of the monitored parameters, which clearly indicated the high efficiency of the refining process. It is also worth mentioning that the efficiency of all three types of rotors at the end of the lifespan is comparable to the results at 25% up to 75% rotor wear;Upon reaching 1100 duty cycles for Type A, Type B and Type C rotors, it can be stated that the components show 100% wear and the rotors need to be replaced to avoid destruction of the graphite components.

It should be noted that in practice, the quality of the melt depends on the balance of customer needs, the time required to process the melt and the price of casts. The results show that it is possible to achieve a better melt quality, but at a price that is economically unacceptable and unnecessary due to customer needs. Thus, in optimizing the degassing process, it is necessary to find a balanced solution which would ensure the required quality in such technological interventions, which will correspond to the set costs of the entire process on the side of the manufacturer.

The obtained results enable mapping the refining process efficiency, in other words the degassing of AlSi9Cu3(Fe) melt during technological processing at 0% up to 100% wear of Type A, Type B and Type C rotors. Based on these results, it will be possible to compare refining process efficiency using not only different rotor type, but it is advisable to pay attention to the material from which they will be made.

## Figures and Tables

**Figure 1 materials-15-04924-f001:**
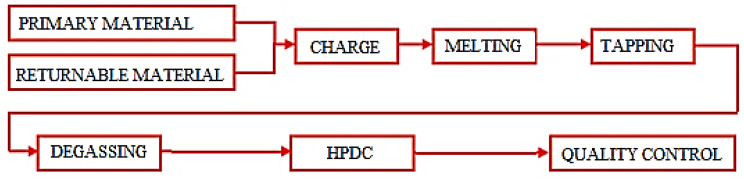
Technological flow of alloy in the production of HPDC cast.

**Figure 2 materials-15-04924-f002:**
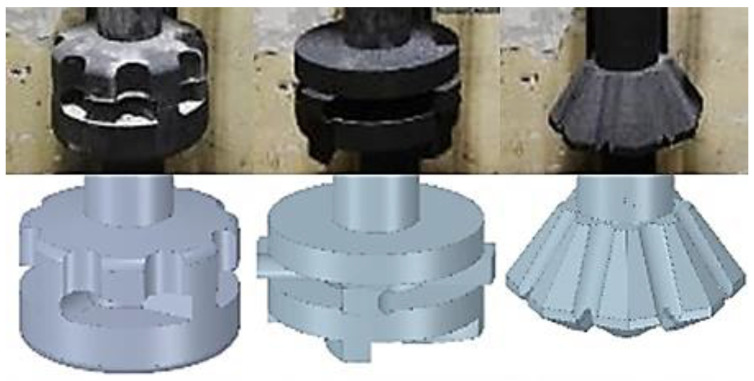
Rotors geometry.

**Figure 3 materials-15-04924-f003:**
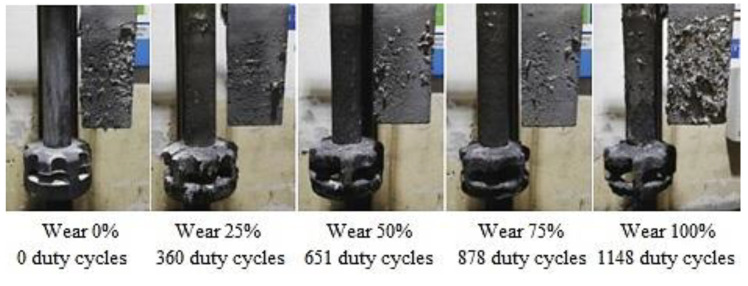
Type A rotor wear.

**Figure 4 materials-15-04924-f004:**
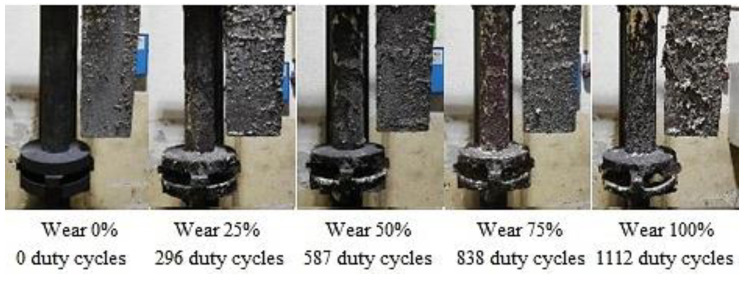
Type B rotor wear.

**Figure 5 materials-15-04924-f005:**
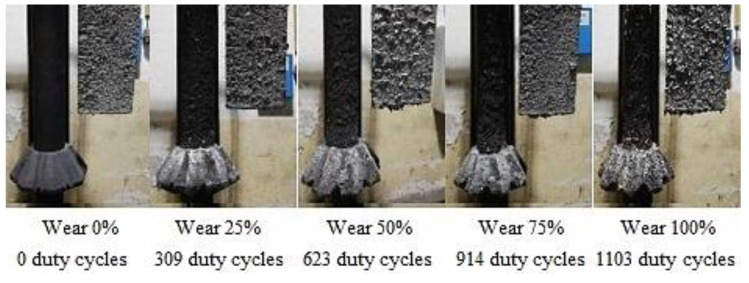
Type C rotor wear.

**Figure 6 materials-15-04924-f006:**
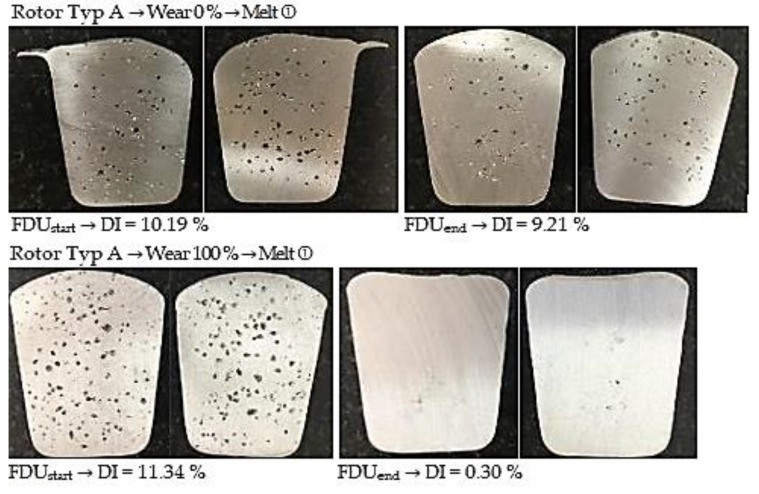
Comparison of degassing at 0% and 100% wear for Type A rotor.

**Figure 7 materials-15-04924-f007:**
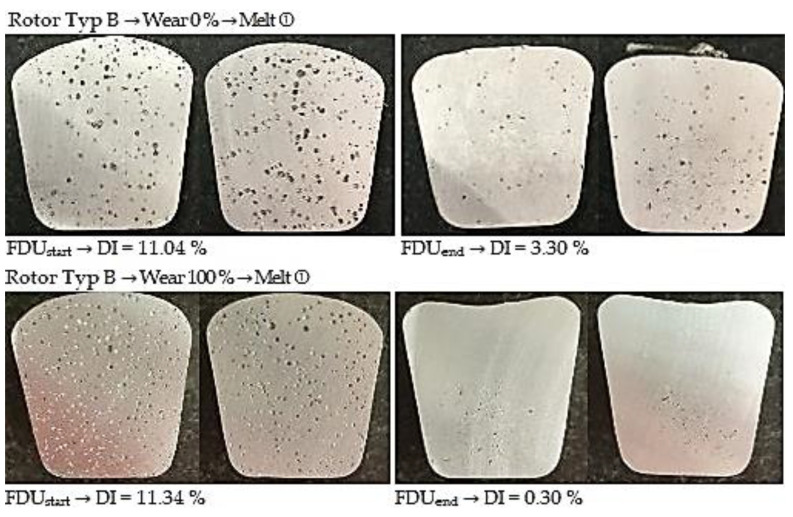
Comparison of degassing at 0% and 100% wear for Type B rotor.

**Figure 8 materials-15-04924-f008:**
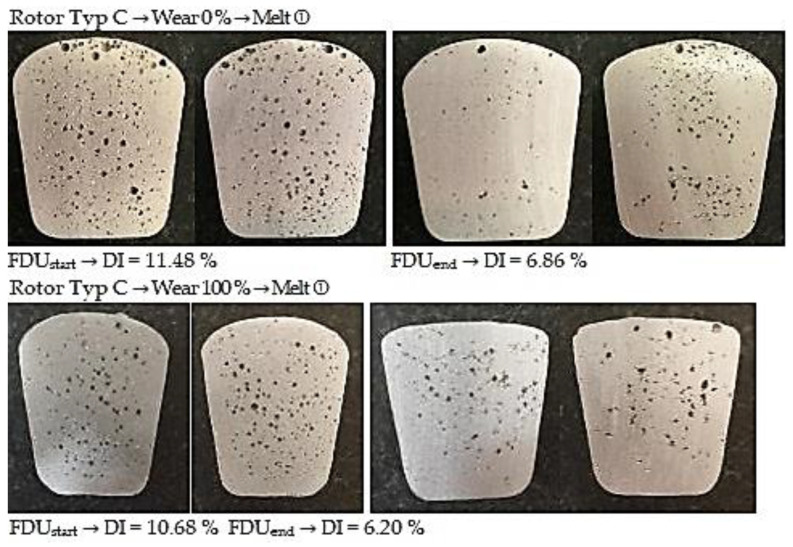
Comparison of degassing at 0% and 100% wear for Type C rotor.

**Figure 9 materials-15-04924-f009:**
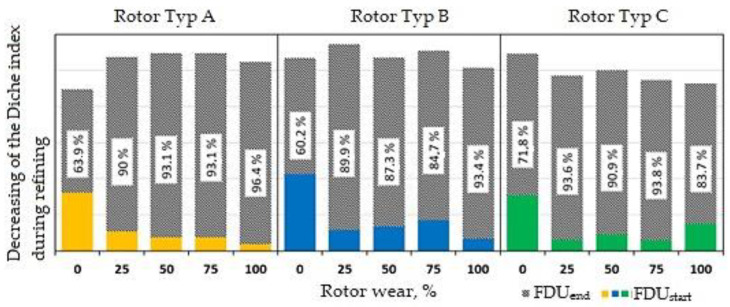
Efficiency comparison of individual types of rotors.

**Table 1 materials-15-04924-t001:** Prescribed chemical composition of EN AC-46000 alloy in accordance with standard EN1706.

Element		wt. % Volume	
		min.	max
Si		8.00	11.00
Fe		---	1.3
Cu		2.00	4.00
Mn		---	0.55
Mg		0.05	0.55
Cr		---	0.15
Ni		---	0.55
Zn		---	1.20
Pb		---	0.35
Sn		---	0.15
Ti		---	0.25
Other ^1, 2^	every	---	0.05
	total	---	0.25
Al		residue	

^1^ “other” does not include modifying or refining elements such as Na, Sr, Sb a P; ^2^ “other” includes all elements not listed in the table or without specified values.

**Table 2 materials-15-04924-t002:** Technological parameters of the refining process.

Parameter	Value
Degassing time, s	180
Rotor speed during rinsing cycle (start x end), n/min	50
Gas flow rate during rinsing cycle (start x end), L/min	5–7
Rotor speed during duty cycle (degassing), n/min	300–350
Gas flow rate during duty cycle (degassing), L/min	15–17

**Table 3 materials-15-04924-t003:** Type A rotor—setting of refining technological parameters.

Rotor	Refining Process Parameters	
	Material	Graphite
	Refining time, s	180
Type A	Revolutions, n/min	350
	Gas flow, L/min	17
	Duty height, mm	160

**Table 4 materials-15-04924-t004:** Type B rotor—setting of refining technological parameters.

Rotor	Refining Process Parameters	
	Material	Graphite
	Refining time, s	180
Type B	Revolutions, n/min	350
	Gas flow, L/min	17
	Duty height, mm	200

**Table 5 materials-15-04924-t005:** Type C rotor—setting of refining technological parameters.

Rotor	Refining Process Parameters	
	Material	Graphite
	Refining time, s	180
Type C	Revolutions, n/min	350
	Gas flow, L/min	17
	Duty height, mm	150

**Table 6 materials-15-04924-t006:** Determined rotor wear values depending on the number of cycles performed.

Rotor Wear	Number of Cycles
0%	0
25%	330 ± 50
50%	650 ± 50
75%	880 ± 50
100%	1100 ± 50

**Table 7 materials-15-04924-t007:** Average chemical composition of samples taken from melts during operational experiments—type A rotor.

	Rotor Wear (Type A)				
	0%	25%	50%	75%	100%
	Number of Samples Tested				
	10	9	8	8	9
Element	Element content, wt. %				
Si	8.702	8.901	8.406	8.698	8.550
Fe	0.935	0.882	0.916	1.059	0.879
Cu	2.153	2.186	2.022	2.115	2.197
Mn	0.340	0.298	0.285	0.283	0.270
Mg	0.207	0.224	0.186	0.183	0.189
Cr	0.071	0.054	0.053	0.082	0.045
Ni	0.100	0.085	0.129	0.134	0.111
Zn	0.676	0.893	0.818	0.741	0.761
Pb	0.064	0.053	0.043	0.041	0.043
Sn	0.024	0.022	0.014	0.016	0.016
Ti	0.049	0.041	0.038	0.038	0.036
Al	86.624	86.311	87.025	86.570	86.862

**Table 8 materials-15-04924-t008:** Average chemical composition of samples taken from melts during operational experiments—type B rotor.

	Rotor Wear (Type B)				
	0%	25%	50%	75%	100%
	Number of Samples Tested				
	11	14	12	11	11
Element	Element Content, wt. %				
Si	8.551	8.929	9.188	8.601	9.235
Fe	0.943	1.020	0.943	0.899	0.888
Cu	2.301	2.322	2.062	2.349	2.176
Mn	0.312	0.284	0.306	0.299	0.294
Mg	0.202	0.198	0.212	0.210	0.206
Cr	0.063	0.050	0.057	0.069	0.062
Ni	0.066	0.068	0.067	0.085	0.102
Zn	0.930	0.849	0.897	0.772	0.613
Pb	0.057	0.048	0.062	0.051	0.042
Sn	0.021	0.020	0.035	0.025	0.017
Ti	0.041	0.040	0.047	0.046	0.046
Al	86.480	86.130	86.080	86.550	86.270

**Table 9 materials-15-04924-t009:** Average chemical composition of samples taken from melts during operational experiments—type C rotor.

	Rotor Wear (Type C)				
	0%	25%	50%	75%	100%
	Number of Samples Tested				
	15	10	15	17	15
Element	Element content, wt. %				
Si	9.188	9.050	8.945	8.926	9.645
Fe	0.862	0.916	0.868	1.010	1.010
Cu	2.409	2.269	2.083	2.074	2.099
Mn	0.261	0.248	0.265	0.269	0.285
Mg	0.232	0.203	0.271	0.219	0.201
Cr	0.060	0.070	0.050	0.073	0.068
Ni	0.089	0.088	0.091	0.056	0.066
Zn	0.941	0.919	0.968	0.810	0.910
Pb	0.047	0.056	0.060	0.056	0.065
Sn	0.028	0.025	0.031	0.020	0.028
Ti	0.048	0.045	0.043	0.045	0.042
Al	85.770	86.060	89.260	86.380	85.51

**Table 10 materials-15-04924-t010:** Statistical summary of measure values of Dichte Index—type A rotor.

Parameter	FDU_start_				
Rotor wear	0%	25%	50%	75%	100%
Expected value	8.96	10.74	10.96	10.97	10.48
Expected value error	0.53	0.37	0.30	0.27	0.28
Median	9.27	10.62	10.86	11.07	10.79
Standard deviation	1.49	1.11	0.85	0.76	0.84
Variance	2.23	1.22	0.72	0.58	0.70
Difference max-min	4.64	3.43	2.73	2.46	2.70
Minimum	5.60	9.24	10.15	9.27	8.64
Maximum	10.24	12.67	12.88	11.73	11.34
Samples taken	8	9	8	8	9
	**FDU_end_**				
Rotor wear	0%	25%	50%	75%	100%
Expected value	3.23	1.08	0.76	0.76	0.38
Expected value error	0.97	0.16	0.12	0.09	0.04
Median	2.12	0.92	0.73	0.72	0.33
Standard deviation	2.73	0.47	0.34	0.24	0.13
Variance	7.47	0.22	0.11	0.06	0.02
Difference max-min	7.64	1.57	1.02	0.78	0.38
Minimum	1.57	0.43	0.28	0.28	0.24
Maximum	9.21	2.00	1.30	1.06	0.62
Samples taken	8	9	8	8	9

**Table 11 materials-15-04924-t011:** Statistical summary of measure values of Dichte Index—type B rotor.

Parameter	FDU_start_				
Rotor wear	0%	25%	50%	75%	100%
Expected value	10.68	11.44	10.71	11.09	10.16
Expected value error	0.26	0.27	0.46	0.29	0.30
Median	10.90	11.79	11.18	10.99	10.37
Standard deviation	0.88	0.98	1.59	0.98	0.94
Variance	0.77	0.96	2.54	0.95	0.88
Difference max-min	3.21	3.15	5.21	2.97	3.01
Minimum	8.60	9.84	7.34	9.60	8.32
Maximum	11.81	12.99	12.55	12.57	11.33
Samples taken	11	13	12	11	11
	**FDU_end_**				
Rotor wear	0%	25%	50%	75%	100%
Expected value	4.25	1.15	1.36	1.69	0.67
Expected value error	0.62	0.16	0.20	0.24	0.08
Median	3.65	1.09	1.33	1.43	0.57
Standard deviation	2.07	0.59	0.69	0.81	0.24
Variance	4.27	0.35	0.47	0.65	0.06
Difference max-min	7.48	1.99	2.69	2.69	0.81
Minimum	1.11	0.26	0.23	0.59	0.33
Maximum	8.59	2.25	2.92	3.28	1.14
Samples taken	11	14	12	11	11

**Table 12 materials-15-04924-t012:** Statistical summary of measure values of Dichte Index—type C rotor.

Parameter	FDU_start_				
Rotor wear	0%	25%	50%	75%	100%
Expected value	10.94	9.71	10.00	9.47	9.29
Expected value error	0.41	0.33	0.32	0.24	0.43
Median	11.14	9.80	10.35	9.56	9.31
Standard deviation	1.58	1.05	1.24	0.95	1.48
Variance	2.49	1.09	1.54	0.90	2.18
Difference max-min	5.54	3.18	4.74	3.21	5.09
Minimum	7.94	7.90	7.08	7.73	6.36
Maximum	11.48	11.08	11.82	10.94	11.45
Samples taken	15	10	15	16	15
	**FDU_end_**				
Rotor wear	0%	25%	50%	75%	100%
Expected value	3.08	0.62	0.91	0.59	1.52
Expected value error	0.99	0.09	0.12	0.07	0.19
Median	1.51	0.70	0.99	0.56	1.49
Standard deviation	3.82	0.24	0.45	0.27	0.66
Variance	14.60	0.06	0.20	0.07	0.44
Difference max-min	14.84	0.61	1.55	0.89	2.13
Minimum	0.68	0.27	0.24	0.19	0.57
Maximum	15.52	0.88	1.79	1.08	2.70
Samples taken	15	7	15	16	15

## Data Availability

Not applicable.

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
