# Peer review of "Aluminum Melt Degassing Process Evaluation Depending on the Design and the Degree of the FDU Unit Graphite Rotor Wear"

_materials, 2022, doi:10.3390/ma15144924_

Round 1

Reviewer 1 Report

The authors have done much work on the rotor design and its influnce on the degrassing process in aluminum alloys. The paper is well structured and informative. There are some questions to address:

1. If the rotor is made of graphite and worn gradually, does it contaminate the molten Al? The elements analysis does not contain C.

2. Are the pores in Figs. 5-7 hydrogen? Do they caused by stirring?

3. Explaination of FDU should be given when it appears at the first time.

4. The figures are not clear to read.

Author Response

Thank you for reviewing the manuscript and your comments.

I would like to respond to your questions and comments:

  1. In principle, it is true, and it follows from the Foundry Theory, that carbon (C) and aluminum (Al) do not react with each other. Aluminum reacts preferentially with elements such as hydrogen (H), oxygen (O). Therefore, the EN 1706 standard does not consider the evaluation of the presence of carbon in the EN AC-46000 alloy.
  2. It is considered that the pores are preferentially caused by hydrogen. However, the presence of other gases is not explicitly excluded.

    The technological parameters of degassing were constant for all types of rotors. The decrease in the volume of gases in the sample after degassing is evident. The effect of mixing the melt during degassing on the volume of gases in the test samples is minor to negligible. Therefore, it is not possible to claim that the presented test samples are not affected by mixing.

  3. It is edited in line 12.
  4. It is edited for better resolution

Reviewer 2 Report

This is a well-written paper and it can be accepted after some minor concerns as follows:

1- in the abstract it was mentioned that "The submitted article addresses the issue of..." this phrase should be changed to "This paper addresses...".

2- in the Tables 4, 5 and 6, the presented images should be removed and only numeric data with definitions should be mentioned. The related images should be added in another Figure after the tables.

3- Figures 5, 6 and 7 should be enlarged.

Author Response

Thank you for reviewing the manuscript. I agree with your comments and am attaching a statement.

  1. Edited in line 11.
  2. Edited as per requirements. Added Figure 2, subsequently edited text.
  3. Edited as per requirements.

Reviewer 3 Report

The manuscript deals with the degasification of the AlSi9Cu3(Fe) alloy using FDU unit. Authors used three different graphite rotors in the experiment. The manuscript is well written and organized. Although the manuscript contains enough experiments, some major revision is needed before it can be considered for publication. The main comments are given below:

1. Tables 7-9 contain average values of chemical composition measured by OES. Since average values are given, tables should also contain standard deviations, giving also an information about reproducibility of samples (or chemical homogeneity across the samples).

2. Authors mention in the conclusions (point 2) that initial moisture of the graphite component can be minimalized. This statement is, however, rather general. Based on the parameters (heating temperature and time) used in this work, authors could at least estimate/concretize how the parameters should be modified to minimize this phenomenon.

3. What does 100% wear correspond to, besides the number of cycles? Can this value be expressed e.g. using dimensional loss of the rotor? This should be specified in the Materials and Methods, if relevant.

4. Between lines 149 and 152, authors mentioned: “The disadvantage of graphite rotors is their lower service life caused by oxidation at higher temperatures (graphite oxidizes from about 500 °C) and also by abrasion. The peeling of graphite particles from the rotor has a negative effect on the purity of the aluminum melt after refining.” The relevant discussion to this based on the experimental results is, however, missing in the manuscript. Did the authors detect any content of C and O in the samples? Please, give a statement.

5. Quality of figures is low. For example, details in Figs. 2-4 are hardly visible. Moreover, the scale in the case of details is missing. The images resolution should be increased.

6. The abbreviation “FDU” should be explained earlier in the manuscript, for instance in line 88.

7. Line 74: a citation is needed.

8. The text in lines 93-94 and 187-188 should be moved to the “Materials and Methods” section. Some further details / parameters are welcomed.

9. The format of References should be uniform. Please, check Refs. 14, 15 and 21.

10. There are also some minor flaws in the manuscript:

- Keywords: keyword aluminium alloy,

- line 66: by ---> be,

- Tables 1, 7-9: wt.% volume ---> rather element content, wt.%,

- unit n/min is a little bit confusing / non-standard, I would rather use either r/min, or min-1.

Author Response

Thank you for reviewing the manuscript.
I agree with your comments and am attaching a statement:

  1. The author's team (which was supplemented by representatives of the company in which the experiments were conducted) understands the reminder and the possible need for secondary statistical variables. On the other hand, we are limited by the framework of the project on the basis of which the article was created and partly by company regulations and the depth of data disclosure.
  2. Comment accepted and included in text - line 329
  3. For the purposes of this experiment, this value is not relevant. The wear of the rotor depending on the cycles and the loss of dimensions and volume will be evaluated in the following works. The evaluation of the loss of volume and the change of geometry depending on the number of cycles will be (and is currently being worked on) evaluated by comparing the geometry of the rotors using 3D touch measuring devices and optical laser scanners.
  4. Oxygen (O) reacts with aluminum (Al) to form Al2O3. The latter has the ability to envelop hydrogen (H) bubbles, thus reducing its solubility and the possibility of exclusion from the melt or from the volume of the final casting. This assumption and fact is confirmed by the higher pore volume in samples made with lower rotor wear.
    Metallographic analysis was performed as part of a series of other experiments using SEM and EDX analyses. The presence of inclusions in the samples and final castings was confirmed. As these experiments were carried out as part of an internal casting quality assessment, the exact specifications cannot be disclosed. The released graphite particles are carried by bubbles of inert gas to the surface of the melt, where they are captured in a layer of covering salt. Its analysis was not performed. The presence of carbon or graphite in the castings was not proven.
  5. It has been edited in line with other reviewers' reviews
  6. It has been edited in line with other reviewers' reviews
  7. is added in line 82
  8. The relevant lines 93-94, now 100 - 101 belong to the clarification of the nature of the article, therefore they are in the Introduction
  9. I'm sorry, but I don't see the mistake. Can you explain it to me more precisely?
  10. Adjusted

All edits in the manuscript are highlighted in red

Round 2

Reviewer 3 Report

Authors satisfactory answered all my questions and comments. Therefore, the manuscript can be considered to be published.

Just a note to my previous comment no. 9 related to the References section: Where relevant, please include the names of all co-authors instead of using "et al.".